# Europe's offshore winds assessed from SAR, ASCAT and WRF

Charlotte B. Hasager[1], Andrea N. Hahmann[1], Tobias Ahsbahs[1], Ioanna Karagali[1], Tija Sile[2], Merete Badger[1], Jakob Mann[1]

[1]Department of Wind Energy, Technical University of Denmark, Frederiksborgvej 399, 4000 Roskilde, Denmark
[2]Department of Physics, University of Latvia, Jelgavas iela 3, Riga, LV-1004, Latvia

*Correspondence to*: Charlotte B. Hasager (cbha@dtu.dk)

**Abstract.** Europe's offshore wind resource mapping is part of the New European Wind Atlas (NEWA) international consortium effort. This study presents the results of analysis of Synthetic Aperture Radar (SAR) ocean wind maps based on Envisat and Sentinel-1 with a brief description of the wind retrieval process and Advanced SCATterometer (ASCAT) ocean wind maps. The wind statistics at 10 m and 100 m above mean sea level (AMSL) height using an extrapolation procedure involving simulated long-term stability over oceans is presented for both SAR and ASCAT. Furthermore, the Weather Research and Forecasting (WRF) offshore wind atlas of NEWA is presented. This has 3 km grid spacing with data every 30 minutes during 30 years from 1989 to 2018, while ASCAT has 12.5 km and SAR has 2 km grid spacing. Offshore mean wind speed maps at 100 m AMSL height from ASCAT, SAR, WRF and ERA5 at a European scale are compared. A case study on offshore winds near Crete compares SAR and WRF for flow from north, west and all directions.

The paper highlights the ability of the WRF model to simulate the overall European wind climatology and the near coastal winds constrained by the resolution of the coastal topography in the WRF model simulations.

## 1 Introduction

The extraction of energy from wind is part of the clean energy transition. It supports society to reach the objectives of the Paris Climate Change agreement and the Sustainable Development Goals. Wind energy in Europe provided 14% of total electricity consumption in 2018. This share will increase in coming years. By the end of 2018, the installed offshore capacity reached 18.5 GW, which is approximately 10% of Europe's total wind energy capacity (Wind Europe, 2019).

Beyond the beneficial impact on reducing carbon dioxide emissions, the offshore wind energy industry is a significant economical factor. According to the Organisation for Economic Co-operation and Development (OECD, 2016), the gross value added of all ocean-based industries globally will double from USD 1.5 in 2010 to 3 trillion by 2030. Offshore wind energy has the highest relative growth rate of the ocean-based industries. In Europe alone, the investments in 2018 in new offshore wind amounted to €10.3bn, a 37% increase from 2017 (Wind Europe, 2019).

Many countries in Europe have operating offshore wind farms. The North Sea accounts for 70% of all installed offshore wind capacity in Europe, followed by the Irish Sea (16%), the Baltic Sea (12%), and the Atlantic Ocean (2%). The longest distance from shore of operating wind turbines exceeds 100 km while permits are given for installation as far as 200 km offshore (Wind Europe, 2019). The expectation is that offshore wind energy will expand to more European seas and that new wind farms are erected in clusters, which already exist in parts of the North Sea (4C Offshore, 2019).

The New European Wind Atlas (NEWA) project focused on experimental campaigns across Europe in different terrain types. These experiments provide unique data for validation of wind models (Petersen *et al.*, 2014; Mann *et al.*, 2017; Witze, 2017). Two of the field experiments are relevant for offshore wind resource mapping. The first is the coastal experiment RUNE with a floating lidar system, three long-range horizontally scanning wind lidars and several vertical wind profiling lidars installed at the North Sea coastline (Floors *et al.*, 2016) nearby the tall meteorological masts at Høvsøre in Denmark (Peña *et al.*, 2015). The second is the wind profiling lidar installed at the ferry link between Kiel and Klaipeda in the Baltic Sea (Gottschall *et al.*, 2018). The two experiments had a duration of around six months. In addition to the dedicated experiments, several years of meteorological observations from tall offshore masts all located in the Northern European Seas are used in preparation of the NEWA offshore wind atlas.

The NEWA project (2015-2019) produced the novel state of the art offshore wind atlas for European Seas covering a minimum distance up to 100 km offshore and the entire North Sea and Baltic Sea, excluding Iceland. In addition to the entire wind atlas simulated using the Weather, Research and Forecasting (WRF) model (Hahmann et al. 2019), also satellite Synthetic Aperture Radar (SAR) and Advanced Scatterometer (ASCAT) ocean winds are processed and analyzed for wind resource assessment.

The overall objective of the study is to present the new European Offshore Wind Atlas and to examine the similarities and differences of wind maps based on ASCAT, SAR and the WRF model. The study focuses on how to use satellite observations for model comparison beyond single cases, and specifically to investigate how different are the 100 m AMSL mean winds based on ASCAT, SAR and WRF.

## 2 Background

In the planning phase of a wind farm project there is need for information on the wind resource (Emeis, 2012; Landberg, 2012; Petersen and Troen, 2012). The methodologies for offshore wind resource assessment rely on wind observations from offshore meteorological masts, wind lidar, SODAR (sound detection and ranging), satellite images and modelling (Sempreviva *et al.*, 2008). The first atlas of the European wind resource covered only land (Troen and Petersen, 1989) and was later extended to offshore (Petersen, 1992). Modelling of wind resources has a long tradition starting with the above-mentioned wind atlas.

Recent offshore model-based wind atlases for the European seas include the German Bight (Jimenez *et al.*, 2006), the Mediterranean Sea (Lavignini *et al.*, 2006), the UK (UK Renewables Atlas, 2008; The Crowne Estate, 2015), the North Sea (Berge *et al.*, 2009), the European Seas (EEA, 2009), the South Baltic Sea (Peña *et al.*, 2011), the Baltic and North Sea (Hahmann *et al.*, 2015) and the Dutch waters (KNMI, 2019).

Offshore wind resource assessment based on *in situ* meteorological wind observations in the Baltic and North Sea (see review in Sempreviva *et al.*, 2008), Italy (Casale *et al.*, 2010) and Malta (Farrugia and Sant, 2016) provide local information. Furthermore, the meteorological observations are useful for comparison to model results to select suitable atmospheric model setup and to assess the model performance (Jimenez *et al.*, 2006; Berge *et al.*, 2009; Hahmann *et al.*, 2015).

Satellite remote sensing used to assess offshore wind resources for the European Seas include scatterometer and SAR measurements. Scatterometer estimates have been validated for the Mediterranean Sea with buoy data (Furevik *et al.*, 2011) and for the Northern European Seas with meteorological mast data (Karagali *et al.*, 2013a; Karagali *et al.*, 2014; Karagali *et al.* 2018a). Soukissian *et al.* (2017) used a blended satellite product based on six different satellites for the Mediterranean Sea and compared to buoy data.

Satellite SAR was used for resource assessment for the North Sea (Hasager *et al.*, 2005; Christiansen *et al.*, 2006; Badger *et al.*, 2010) and the Baltic Sea (Hasager *et al.*, 2011; Badger *et al.*, 2016) and was compared to meteorological mast data. Coastal mast data and mesoscale model results were compared to SAR-based wind resource estimates for the Icelandic waters, (Hasager *et al.*, 2015a). Scatterometer data (ASCAT) was also compared to WRF mesoscale model results in the entire European Seas (Karagali *et al.* 2018a, 2018b).

There is potential to also compare model results and satellite data to wind profiling lidar (light detection and ranging) data at offshore platforms (Hasager *et al.*, 2013) and floating wind profile lidar systems (OWA 2018; Bischoff *et al.*, 2018). These are local point data similar to buoy data and meteorological mast data. Recently, new technological advancements provide opportunities for horizontal spatial data comparison, e.g. the Dual-Doppler radar (Nygaard and Newcombe, 2018; Valldecabres *et al.* 2019). Three other spatial data types are horizontally scanning lidar, long row of turbines providing SCADA (Supervisory Control And Data Acquisition) data, and ship-mounted vertical profiling lidar.

Recently, offshore winds observed with long-range scanning lidar at a coastal site at the North Sea (Floors *et al.*, 2016) were compared to SAR winds and showed good comparison within 2 to 5 km from the North Sea coastline. The good agreement was unexpected because the Geophysical Model Function (GMF) used to retrieve winds from SAR is valid in open-ocean and not near the coast. The conclusion of the study is that SAR winds are mapped well as close as 2 km from the coastline at the site investigated (Ahsbahs *et al.*, 2017). Documentation at more complex coastline remains open.

Another recent study found that the SAR-based winds compare slightly better than mesoscale model results to the wind speed observed at 20km long row of turbines. The turbines are operating in an area with a strong horizontal wind gradient along the coast (Ahsbahs *et al.*, 2018).

The third novel spatial comparison method was based on a vertical profiling lidar installed on-board a ferry sailing daily across the Baltic Sea for several hundred kilometers; measurements compared well to mesoscale model results (Gottschall *et al.*, 2018). Data near the harbors were excluded from the analysis. The WRF mesoscale model results generally are better offshore than near coastlines due to the differences between land and sea influencing the atmospheric flow (Hahmann *et al.*, 2010; Hahmann *et al.*, 2015; Floors *et al.*, 2018). The flow is more complex near the coastline than further offshore and fine scale structures may prevail such as land-sea breeze, not resolved by the model.

The presentation of methodology for wind mapping based on ASCAT, SAR and WRF is given in Section 3. Section 4 presents the results for the entire European Seas from ASCAT, SAR and WRF, their inter-comparisons and cross-comparison to ERA5. Section 5 is a case study of offshore winds around Western Crete using SAR and WRF; thus, provides insight to specific details on the two types of data. Section 6 covers a discussion and perspectives regarding the results, followed by conclusions in Section 7.

## 3 Methodology

### 3.1 Area of interest and time period

The offshore part of NEWA covers the European Union, associated states, and Turkey from the coastline and at least 100 km offshore. For the WRF model, the simulations are done for 10 separate subdomains and later merged to provide one unified atlas (Figure 1). The WRF modelling covers 30 years from 1989 to 2018. For the satellite data collection, processing and analysis, it is convenient to select an area of interest within latitudes (here 33.5° to 72.2°) and longitudes (here 19.4°W to 47°E).

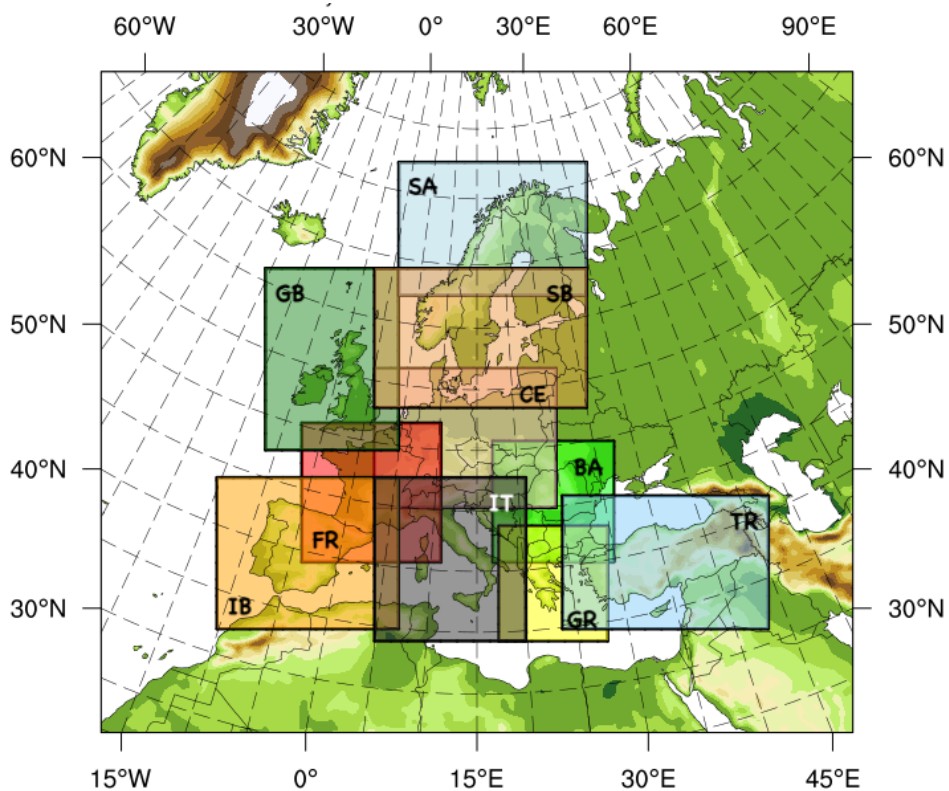

**Figure 1: The WRF modelling domain with 10 subdomains used in the production of the New European Wind Atlas.**

### 3.2 ASCAT and SAR ocean winds processing

The scatterometer ASCAT is on-board the meteorological MetOp-A and B satellites observing from 2007 and 2012, to present, respectively. Metop-C was launched in 2018 although its data are not used in the present study. All are operated by the European Organisation for the Exploitation of Meteorological Satellites (EUMETSAT). The Level 3 data obtained through the Copernicus Marine Environmental Monitoring Service is the Coastal Stress Equivalent Wind product (WIND_GLO_WIND_L3_NRT_OBSERVATIONS_012_002). It includes wind speed and wind direction at 10m height

above sea level at spatial resolution of 12.5 km (de Kloe *et al.*, 2017; CMEMS 2019). Depending on the area of interest, satellite overpass times can range from two to four per day, while the measurements are considered instantaneous (CMEMS-OSI-PUM-012-002). Near coastlines, quality control omits pixels contaminated by land that cause fundamentally different scattering than ocean waves.

Level 1 Wide Swath Mode (WSM) acquisitions from the Envisat ASAR (Advanced SAR) mission, from 2002 to 2012, are collected in its entirety for the area of interest. The scenes used in this study include co-polarized VV and HH scenes (VV is

vertical receive, vertical transmit and HH for horizontal receive and transmit). Envisat was a research mission of the European Space Agency (ESA).

Level 1 Extra Wide (EW) and Interferometric Wide (IW) mode acquisitions from the Sentinel-1A mission (2014-present) and Sentinel-1B (2016-present) are collected in its entirety for the area of interest. The scenes used in this study include EW and IW mode and VV and HH polarization. Sentinel-1A/B are parts of Copernicus, the European Commission's monitoring program. Table 1 lists the source data from ASCAT, Envisat and Sentinel-1.

ASCAT, Envisat and Sentinel-1 are polar orbiting satellites. The number of samples of ocean wind data in any pixel (grid cell) depend upon the data recordings during time and space. For Envisat this was inhomogeneous due to various research priorities in the beginning of the mission. During later years (2008 to 2012), recording was high and consistent in the area of interest. ASCAT-A/B and Sentinel-1A/B are operational monitoring satellites and have frequent coverage in the entire domain since launch. For all satellites, there are more samples available at higher latitudes due to the polar-orbital paths.


**Table 2: List of source data for the European Seas between 1989 and 2018 for ASCAT, SAR and WRF.**

| Source | Mode | Polarization | Swath width (km) | Grid cell (km) | Period (years) |
|---|---|---|---|---|---|
| Envisat | WSM | VV | 405 | 2 | 2002-2012 |
|  |  | HH | 405 | 2 |  |
| Sentinel-1A | IW | VV | 250 | 2 | 2014-2018 |
|  | EW | HH | 400 | 2 |  |
| Sentinel-1B | IW | VV | 250 | 2 | 2016-2018 |
|  | EW | HH | 400 | 2 |  |
| ASCAT-A |  | VV | 500 | 12.5 | 2007-2018 |
| ASCAT-B |  | VV | 500 | 12.5 | 2012-2018 |
| WRF |  |  |  | 3 | 1989-2018 |

The SAR wind retrieval is based upon calibrated radar backscatter values (the Normalized Radar Cross Section) and application of the GMF CMOD5.N (Hersbach, 2010). CMOD5.N gives the equivalent neutral wind at 10m height above sea level. This is

for radar data in VV polarization. There is no GMF for HH data, therefore a conversion, the so-called polarization ratio linking the VV and HH data together needs to be applied before wind retrieval. For HH data, the polarization ratio of Mouche *et al.* (2005) is selected. The *a priori* wind directions needed to perform wind retrieval, are selected at 10m height from the NCEP/NCAR Climate Forecast System Reanalysis (CFSR) reanalysis data until 2010 and the Global Forecast System (GFS) data from 2011 onward. To match the SAR images, an interpolation of wind directions is performed. The SAR Ocean Products

System (SAROPS) software from Johns Hopkins University Applied Physics Laboratory and National Ocean and Atmosphere Agency (JHU APL and NOAA) is used for the processing (Monaldo *et al.*, 2014), which occurs operationally at DTU Wind Energy; all wind retrievals are openly available through https://satwinds.windenergy.dtu.dk/. In regions with sea ice, ocean winds cannot be retrieved and thus these areas are masked out using the National Ice Center's Interactive Multi-sensor Snow and Ice Mapping System (IMS) with daily data at 4 km resolution (National Ice Center, 2008).


Satellite winds retrieved at 10m height are averaged into wind resource statistics using the software for SAR-based wind resource assessment (Hasager *et al.*, 2008, Hasager *et al.*, 2011, Ahsbahs *et al.*, 2019) and for ASCAT using the methodology presented in (Karagali *et al.*, 2018b). Wind turbines offshore have hub heights at around 100m height. Therefore, an extrapolation of wind speed from 10m to 100m height is applied. Previous investigations show that applying a long-term stability correction is superior to neutral logarithmic wind profile in the Baltic Sea (Badger *et al.*, 2016) and in the North Sea (Karagali *et al.*, 2018a). For the NEWA offshore wind atlas, the extrapolation is done similar to Karagali *et al.* (2018a, 2018b) using 10-years of WRF model simulations from Nuño Martinez *et al.* (2018) for the long-term stability correction.

### 3.3 Mesoscale modelling

The WRF model (Skamarock *et al.*, 2008) used for the production run of the New European Wind Atlas is a limited area weather forecast model. The WRF model is a public domain, open-source modelling system, which has previously been used to produce wind atlas for South Africa (Hahmann *et al.*, 2014), the North Sea and Baltic Sea (Hahmann *et al.*, 2015), Denmark (Peña and Hahmann, 2017) and wind statistics for Europe (Nuño Martines *et al.*, 2018).

The production run for NEWA (see Figure 1) was computed on the HPC cluster MareNostrum at the Barcelona Supercomputing Center and on HPC Cluster EDDY at the University of Oldenburg. In order to determine optimal model scheme and forcing, surface input and land surface model, a series of sensitivity tests were conducted and compared to tall meteorological mast data masts and lidar data in northern Europe and the North Sea. No setting was optimal for all, so a compromise was taken, which provided the best verification statistics (see Witha *et al.*, 2019 for more details). In brief, the production run was set up for 10 separate WRF domains, which shared the same outer domain and map projection, and later merged provide one unified atlas (https://map.neweuropeanwindatlas.eu). The WRF model used was a modified version of 3.8.1, setup with the MYNN Planetary Boundary Layer (Nakanishi and Niino, 2009) and Monin-Obukhov surface layer (Monin and Obukhov, 1954) schemes. Forcing for the simulations was from ERA5 reanalysis (ERA5, 2017) at 0.3° x 0.3° grid spacing and OSTIA Sea Surface Temperature (Donlon *et al.* 2012) at 1/20° grid spacing. The CORINE land cover data at 100 m resolution was used to define the land use classes (Copernicus Land Monitoring Service, 2019), except for areas it does not cover, then ESA CCI data is used. The NOAH land surface model and icing WSM5 plus ice code and sum of cloud and ice humidity are used. The WRF simulations used three nested domains at 27 km, 9 km and 3 km and 61 vertical layers, with 8-

day overlapping runs using spectral nudging with 24-hour spin-up (see Hahmann *et al.,* 2015 for details on the technique). There are 20 model levels below 1 km, the lowest levels are located at 5.6, 21.8, 40.4, 56.6, 72.8, 90.7, 113.2, 140.1, 170.7,
205.3, 244.5 m above ground level. The years covered and spatial resolution are listed in Table 1.

## 4 Offshore wind speed assessment for Europe

### 4.1 Satellite-based offshore wind speed maps

Figure 2 shows the offshore wind speed maps for the European Seas based on the entire archive of ASCAT at 10m and 100m height, the number of samples and wind speed difference at 100m using extrapolation with long-term stability correction minus neutral profile extrapolation. Similar results for SAR are shown in Figure 3.


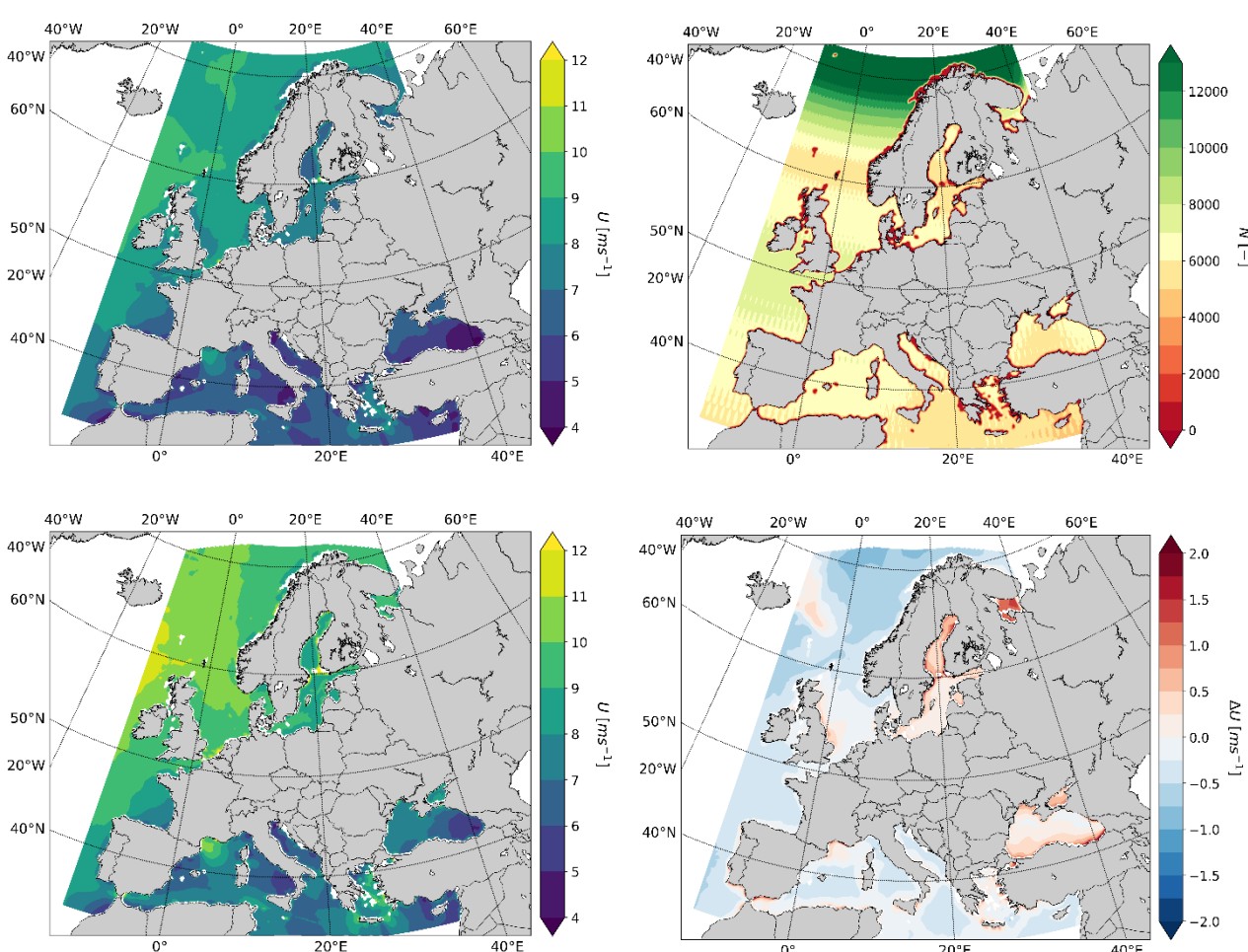

**Figure 2: ASCAT: Mean wind speed (m s⁻¹) at 10 m height (top left), number of samples (top right), mean wind speed at 100m including long-term stability correction for extrapolation (bottom left) and difference in wind speed at 100 m AMSL height based on long-term stability correction minus neutral wind profile assumption (bottom right).**


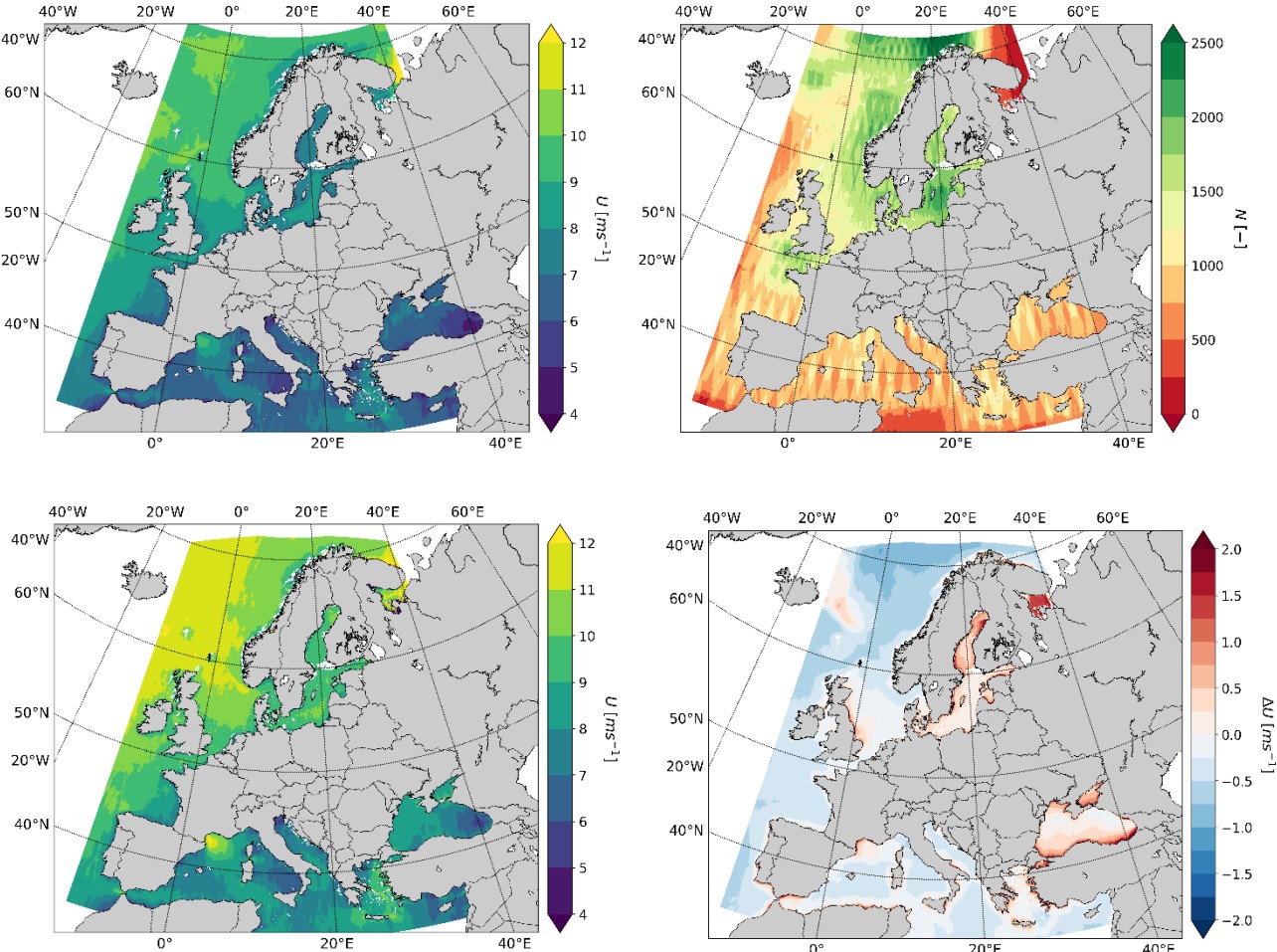

**Figure 3: Envisat ASAR and Sentinel-1 combined: Mean wind speed (m s⁻¹) at 10m height (top left), number of samples (top right),**
**mean wind speed at 100 m AMSL including long-term stability correction for extrapolation (bottom left) and difference on wind speed at 100 m AMSL height based on long-term stability correction minus neutral wind profile assumption (bottom right).**

The same color scale is used for ASCAT and SAR in Figures 2 and 3, except for the number of samples due to the difference in sample maxima between ASCAT and SAR. The polar orbits result in more frequent sampling at higher latitudes. The
harlequin pattern in sampling is due to the ascending and descending orbits for both ASCAT and SAR but most noticeable for SAR due to the swaths and orbital settings.

For the European Seas, the number of samples in the grid cells for ASCAT is greater than 4,000 and in most places greater than 6,000, up to more than 12,000 at high latitudes (see Figure 2). The number of samples for SAR is between 500 and 2,500 (see Figure 3). For the WRF model, the number is constant at all locations covered with 525,912 samples (every 30 minutes from 1989 to 2018).

The mean wind speed consistently shows higher values for the 100m height than 10m height both in ASCAT and SAR. The wind speed difference maps at 100m based on long-term stability correction minus neutral wind profile assumption shows very similar spatial patterns between ASCAT and SAR, as expected. The variation is up to $\pm 2$ m s$^{-1}$ with high positive values in the Baltic Sea and Black Sea and with high negative values in the Norwegian Sea. Positive values occur for stable conditions. The continental climate dominating the flow in the Baltic Sea and the Black Sea cause the variations. Negative values occur for unstable conditions prevalent in the Norwegian Sea and in the Mediterranean Sea. According to Kara *et al.* (2009) the overall stability in the Mediterranean Sea is slightly unstable. In the North Sea, a gradient is observed with slightly negative values along the continental coast and positive values along the UK coast. This corresponds well with the average stability over the North Sea (Peña and Hahmann, 2012), where unstable conditions prevail along the continental coast and stable conditions near the UK. The Mediterranean Sea has mixed wind speed difference variations dominated by moderately negative values in the central part and positive values in the Greek archipelago and the French Riviera.

**4.2 WRF offshore wind speed map**

The long-term offshore wind speed map at 100m height in the European Seas based on the WRF production run is shown in Figure 4, using the same colour scale as for ASCAT and SAR in Figures 2 and 3.

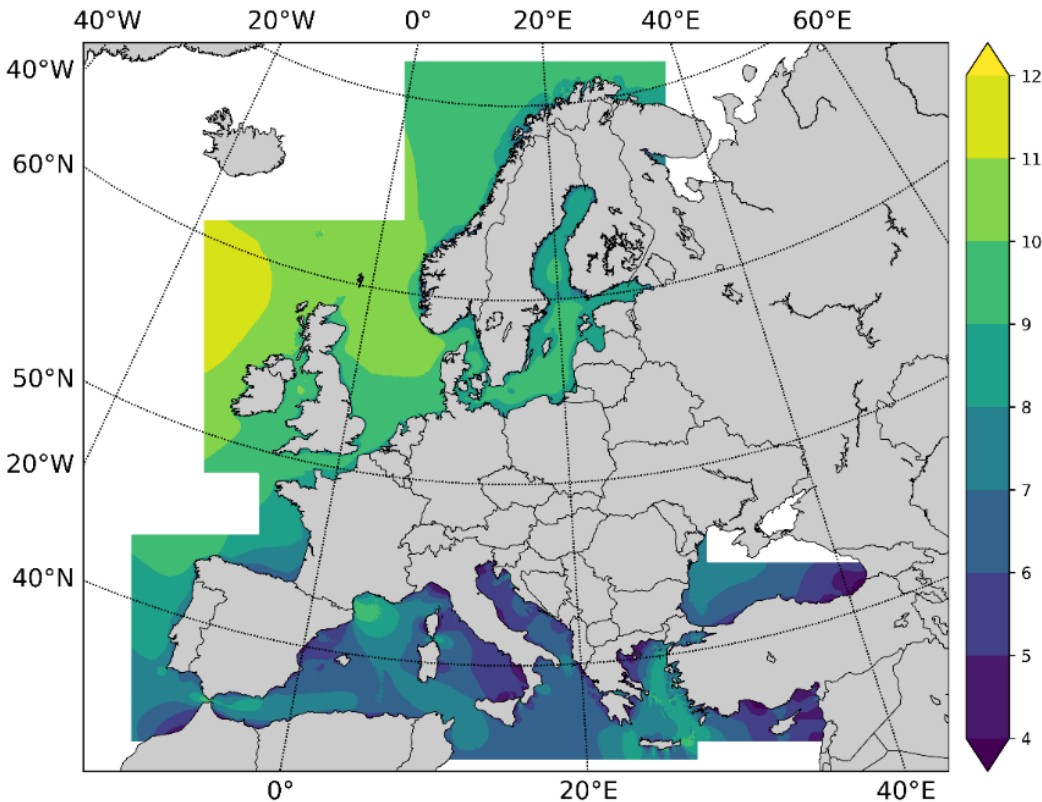

**Figure 4: WRF New European Wind Atlas production run mean wind speed (m s⁻¹) at 100m height for 1989 to 2018 with 3 km spatial grid spacing.**

ASCAT and WRF have many similarities in the spatial wind speed patterns and the range of mean wind speeds at 100m height. The SAR mean wind speed at 100m height appears to be higher than ASCAT and WRF. Furthermore, SAR shows more fine-scale spatial variations than both ASCAT and WRF.

## 4.3 Comparison of offshore mean wind speed maps at 100m height

Comparisons of the ASCAT, SAR and WRF mean wind speed maps at 100m height performed using the long-term stability corrected versions from ASCAT and SAR are shown in Figure 5. ASCAT versus WRF (top left panel) shows lower differences in mean wind speed than SAR versus WRF (top right panel). ASCAT minus SAR (bottom panel) shows a consistent negative bias of winds from SAR, except for some artefact in ASCAT near the Dutch coastline, attributed to higher backscatter from the surface due to the congestion of large ships to and from Rotterdam (notice distinct yellow area in Figure 2 that without ships would be green, in other words, the high backscatter translate into falsely high wind speed).

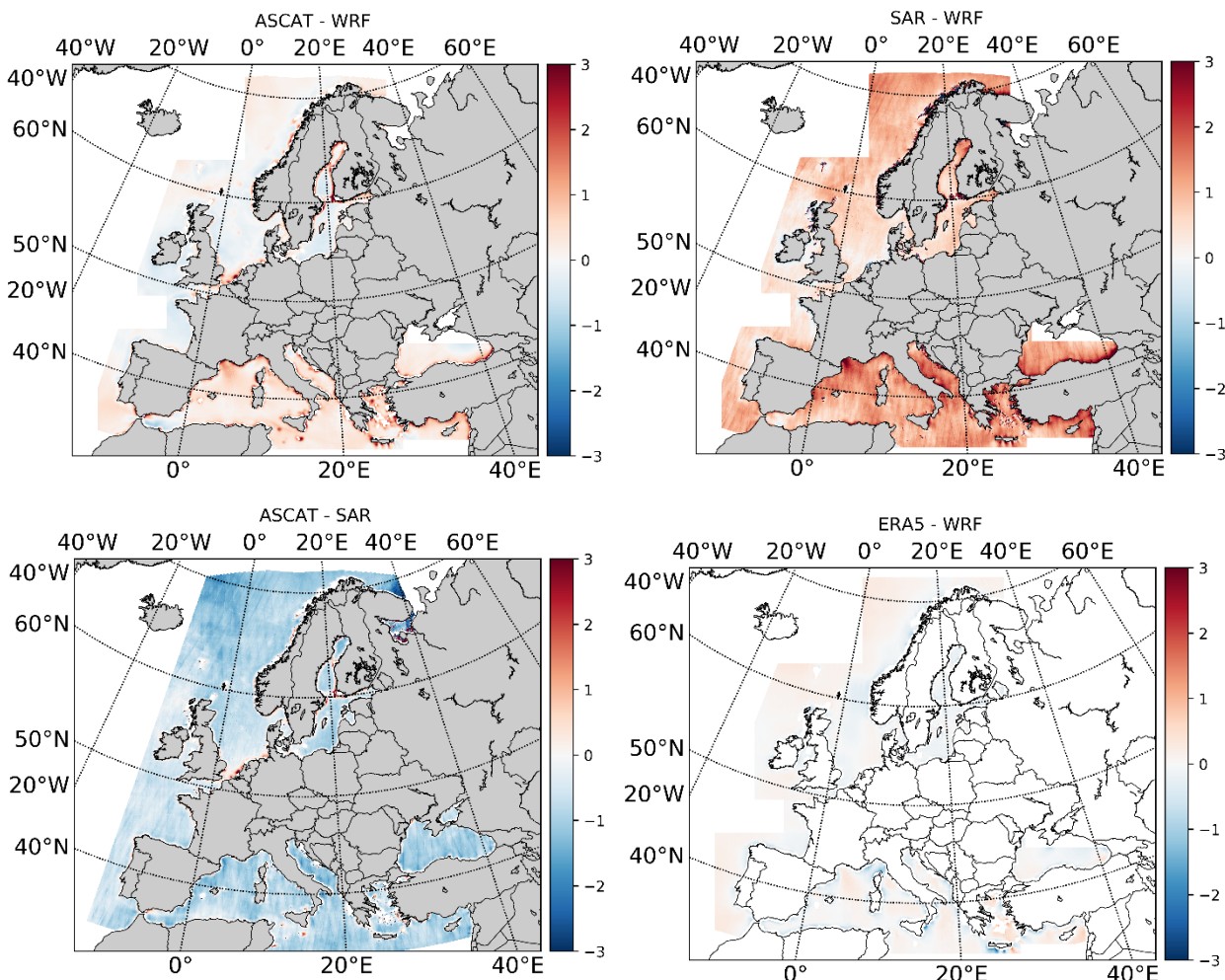

**Figure 5: Comparison of mean wind speed (m s⁻¹) at 100m height: ASCAT minus WRF (top left), SAR minus WRF (top right), ASCAT minus SAR (bottom left) ERA5 minus WRF (bottom right).**

260

The ERA5 mean wind speed at 100m height is included for comparison with WRF in Figure 5 (bottom right). Only grid cells with more than 95% sea according to the ERA5 land mask is included. The mean wind speed difference map of ERA5 minus WRF shows some variations. There are both large positive and large negative values in the Mediterranean Sea. The differences are smaller in the Northern European Seas. Along several coastlines such as the Norwegian Sea, the Atlantic Sea and the

265 Mediterranean Seas large differences are found between the two datasets. These are attributed to the lack of ability in ERA5 to properly resolve the coastal atmospheric flow phenomena such as land-sea breeze and flow intensification. See Beal *et al.* 2015 for further details on coastal atmospheric flow phenomena observed in SAR. ERA5 has a coarse spatial resolution omitting islands such as Bornholm and Isle of Man. Coastal atmospheric flow phenomena near Crete are investigated in section 5

270

Please note the number of samples and the grid spacing are different. WRF has 30-minute values from 30-years (525,912 samples) with 3 km resolution. ERA5 has hourly values from 30 years (262,956 samples) with about 27 km resolution.

From the spatial resolution perspective, it is obvious that SAR resolves finer spatial detail than other products. From spectral analysis of SAR vs. scatterometer winds, it was found that SAR resolves around 4 km features and scatterometer around 25 km features (Karagali *et al.*, 2013b). The latter is comparable in scale to what the WRF model at 3 km grid spacing resolves, i.e. around 20 km (Skamarock, 2004). ERA5 resolves scales around 150 km. The use of structure functions and spectra from other numerical weather models have shown the effective model resolution to be of the order six times lower than the size of the grid cell (Frehlich and Sharman, 2008).

The wind-speed difference error distributions between wind speed at 100m height for ASCAT minus WRF, SAR minus WRF, ASCAT minus SAR and ERA5 minus WRF are shown in Figure 6. ASCAT minus WRF has slightly positive bias and narrow range. ASCAT minus SAR has negative bias and moderate range. SAR minus WRF has positive bias and broad range. The narrow range is expected for products that resolve similar length scales while broader ranges are expected for products that resolve different length scales. The results shown in Figure 6 supports this very well as ASCAT and WRF resolve similar scales and SAR and WRF resolve very different scales. SAR generally show higher wind speeds than ASCAT and WRF. The long positive tails of ASCAT minus WRF and SAR minus WRF are explained by coastal winds in the Mediterranean Sea. ERA5 minus WRF has slightly positive bias and narrow range. All probability density distributions except ASCAT minus SAR, show bi-modal distributions explained by the differences in area of the Northern Seas and Mediterranean Sea, see Figure 5.

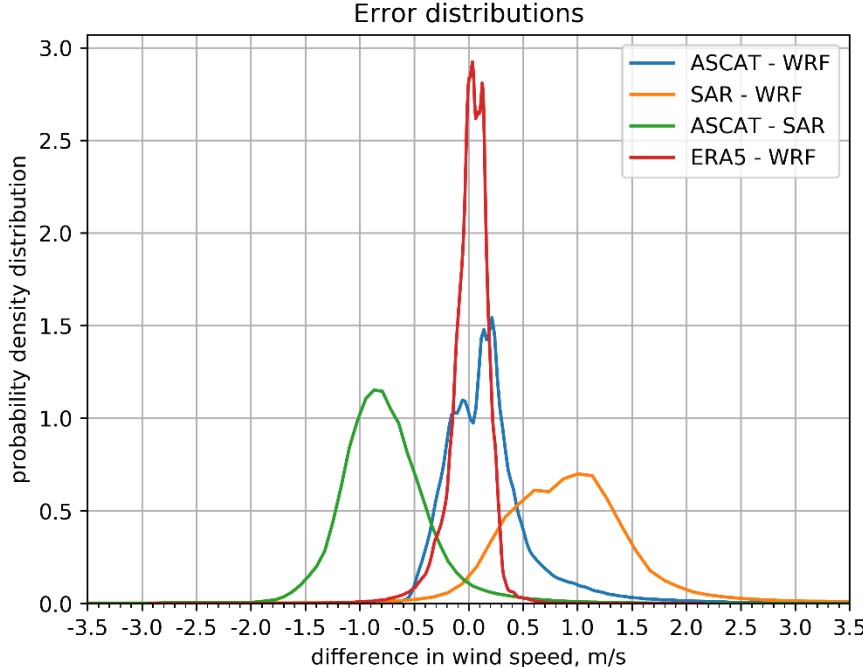

**Figure 6: Error distribution of the wind speed difference at 100 m AMSL for ASCAT minus WRF, SAR minus WRF, ASCAT minus SAR and ERA5-WRF. The distributions are normalized so the area below the curves sums to one.**

## 5 Crete case study

### 5.1 Motivation and aim

The motivation for presenting a case study is two-fold. Firstly, by looking into a small area of interest, spatial details in winds observed can be analysed and used as example for characterizing the SAR and WRF data sources. More specifically, the goal of this case study is to study the interaction between large-scale flow and orography. Secondly, to stimulate interest for further investigation using the different data sources at other locations in Europe and outline the methodology.

### 5.2 Selection of data

The sea surrounding Western Crete is chosen due to interesting mesoscale flow patterns. Figures 7, 8, 10 and 11 show spatial wind patterns in the area. The area of investigation is located between 23.4° to 24.8°E and 34.6° to 36.0°N.  The SAR scenes available from the database satwinds.dtu.dk at DTU Wind Energy are selected. To have spatial consistency between SAR and WRF, only SAR scenes that fully cover the area (consecutive scenes are merged) are selected.

There are 549 SAR scenes between 2002 and 2018 in total from Envisat and Sentinel-1. Only coinciding WRF data are selected. The SAR and WRF mean wind speeds at 10m height are displayed in Figure 7. Some wind features are similar in
SAR and WRF, e.g. lower wind speeds south and north of Crete close to the shore. A distinct jet south of the island is much more pronounced in the WRF data than in SAR. Figure 7 (left panel) shows the height contour lines from the elevation map used in the WRF model. To characterize the complex landscape in Crete, a more detailed elevation map is embedded in the SAR map in Figure 7 (right panel). Small-scale elevation features not represented in the WRF model may explain wind speed differences between WRF and SAR. The jet could be weaker or absent since the fine-scale elevation features, neglected in
WRF, block the atmospheric flow. For instance, what is a simple valley without no obstacles in WRF orography, in reality (and therefore in SAR data) could be blocked by a small mountain range. Koletsis *et al.* (2010) demonstrated the sensitivity of gap wind speeds in a mesoscale model to the changes in the elevation.

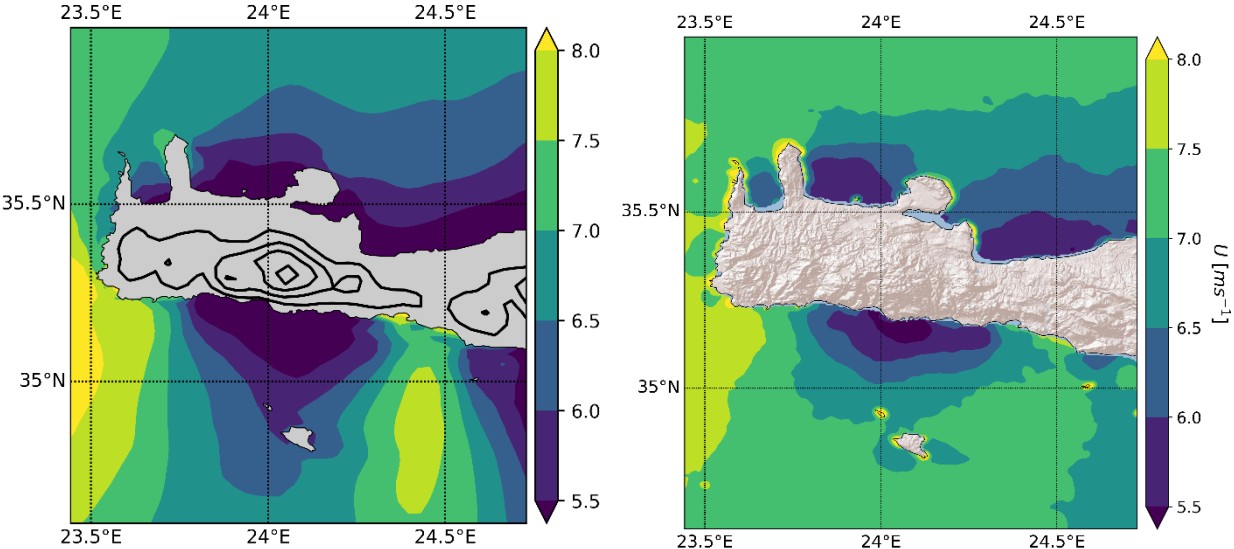

**Figure 7: Mean wind speed (m s$^{-1}$; 2002-2018) at 10 m AMSL. Left panel: WRF. Right panel: SAR. All 549 coinciding scenes are used in both datasets.**

The variability in coastal flow around Crete depends highly on the wind direction. Two of the prevalent wind directions are of particular interest - namely northerly and westerly winds. Northerly winds over Crete are associated with the so-called Etesian
wind often present in the region during summer and known to produce gap flows between the two large mountains in the East side of the island (Lefka Ori) and centre (Idi) (Koletsis *et al.*, 2010). Westerly winds in this region have been associated with trapped lee waves (Miglietta *et al.*, 2013).

As already stated, the goal of this case study is to demonstrate the interaction between large-scale flow and orography. It is

necessary to choose situations where the upwind flow conditions are simple. This is to avoid wind conditions such as low wind speed with poorly defined direction, anti-cyclonic situations and local flows, e.g. sea breezes that could create a complicated wind field that would be difficult to interpret. Therefore, the wind speeds should be sufficiently high, and the wind direction should be representative for the entire domain.

To determine a representative flow direction, ERA5 wind speeds and directions extracted at the locations indicated in Figure 8 are used. Figure 8 also shows the mean wind speed from the 549 coinciding ERA5 model simulations. ERA5 resolves the mean wind speed with much less spatial detail than WRF (compare Figure 8 and Figure 7 (left panel)). The average wind speed at three points (A, C, E) is required to be above 3 m s$^{-1}$. For the wind direction, the centre location upstream (B for northerly, D for westerly) should be within 30° of that direction. We further require that the neighbouring upstream points do

not differ by more than 20° from the centre. Figure 9 illustrates the flow chart used for classification.

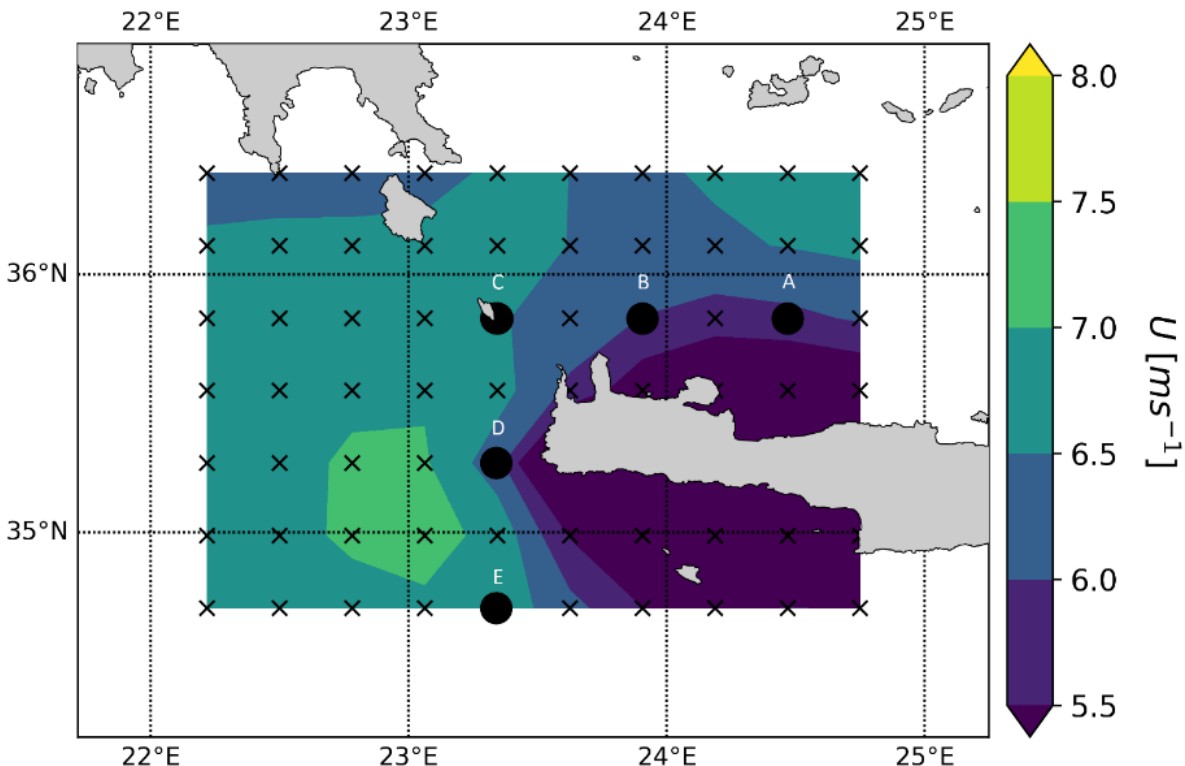

**Figure 8: Mean wind speed (m s$^{-1}$) from ERA5 for 549 cases collocated with SAR scenes around Crete with points used for extracting wind speed and direction for classification. The five locations A to E are mentioned in the text.**

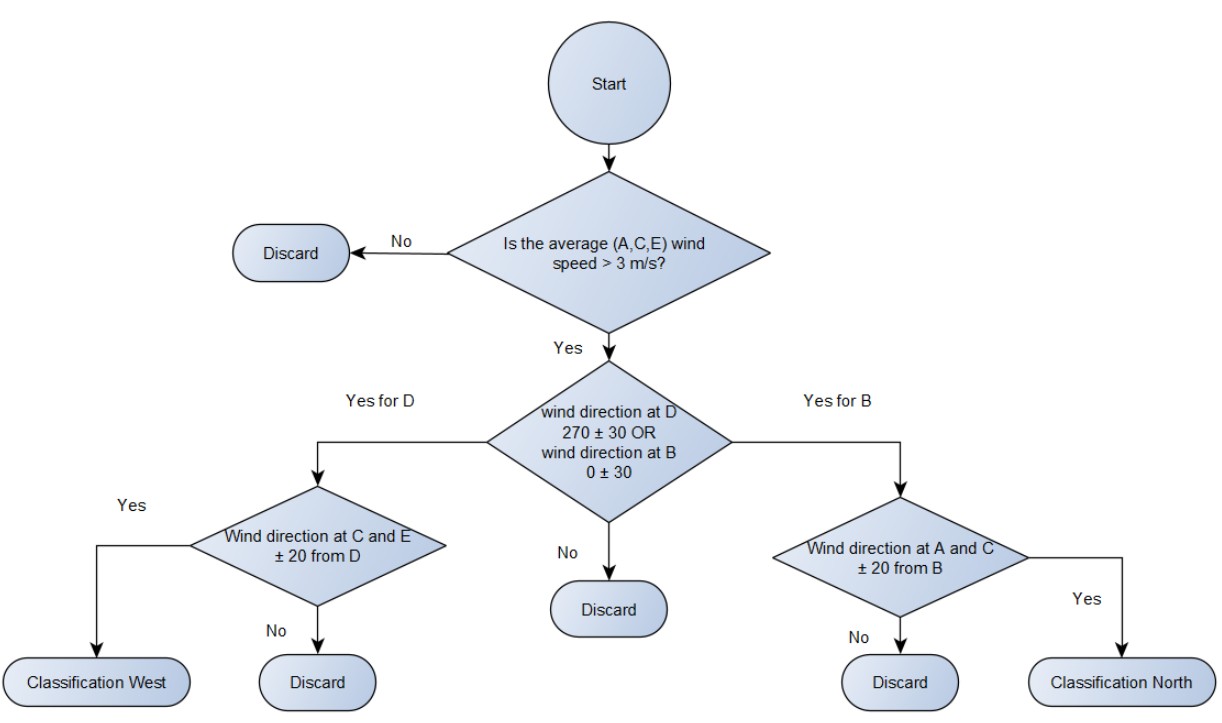

**Figure 9: Flow chart for selection of cases classified as winds from northern and western directions, respectively.**

The mean wind speed maps based on SAR and WRF for 59 cases of northerly and 57 cases of westerly flows are presented in Figure 10. For northerly flow, notable differences exist between the SAR and WRF maps. The WRF winds show strong
shadowing at 24°E and a pronounced jet-like structure at 24.5°E. These features are present in the SAR as well, but much less pronounced. For westerly flow, good agreement between SAR and WRF is noted. Areas of increased wind speed to the south and the north are visible in both maps. A stagnation point area of low wind speed is located on the western side of Crete in both SAR and WRF maps. Stronger winds may increase surface water mixing causing colder sea surface temperature (not shown).


To clarify further similarities and discrepancies between SAR and WRF, two individual examples are chosen and compared to WRF, see Figure 11. One case of northerly flow from 5 July 2017 at 04:24 UTC and one case of westerly flow from 6 May 2017 at 04:24 UTC.

The northerly flow SAR case (Figure 11, top right panel) contains significant atmospheric waves. Although some evidence for atmospheric wave activity is also identified in the WRF data, namely, periodic changes of flow over time (not pictured), no

wave structure similar in wavelength to the one visible in SAR is identified in WRF. This could be because topography of the appropriate resolution is not present in the WRF model and thus cannot be resolved in the mode solution. In addition, the WRF model grid is too coarse to resolve the scale of the observed waves. The wind speed maxima in SAR and WRF compare well

as do the minima for the northerly flow case (Figure 11, top panels). The much weaker jet identified in the average of northerly flows in SAR vs. WRF (in Figure 11, top panels) can, in part, be explained from the gravity wave amplitude maxima not occurring at the same location. The averaging of out-of-phase waves can lead to destructive interference and the result in the average sum of lower values in SAR. In contrast, WRF appears to simulate only the maxima of the non-resolved waves.

The westerly flow case for SAR and WRF (Figure 11, lower panels) also shows atmospheric wave activity. However, in this case the waves seem to have a longer wavelength, and therefore could - although imperfectly - be reproduced in WRF; WRF still showing significantly longer wavelengths than SAR.

In summary, the orography in WRF is somewhat simplified and therefore significant features of critical role in complicated

flows are omitted. Another direction of investigation is to assess whether the atmospheric stability in the flow before it arrives at the obstacle has the correct representation in WRF. Meteorological observations are unfortunately not available for comparison.

The atmospheric stability from WRF has been investigated. We use the stability classes derived from the Obukhov length

following Gryning *et al.* (2007) and map the stability at the time of the two cases shown in Figure 11. The results are presented in Figure 12. For the case with winds from the north, the inflow is very unstable at the northern shore of Crete, but in the high-speed jet south of Crete the flow is neutral. In the low-speed zone south of Crete there is a complicated pattern of atmospheric stability – with a large zone of very unstable flow. For the case with winds from the west, the inflow (West of Crete) is again very unstable, and the wind speed maxima are again associated with neutral stability conditions, but this time in the low wind

speed zones near the coastline north and south of Crete very stable stratification is found.

The results in Figure 12 show a snapshot of stability in time, however, the focus on this paper is the climatological properties. Therefore, the next question arises – are these stability patterns typical. To answer this question, we analyze the stability patterns in the WRF dataset that coincides with SAR scenes (Figure 10). The stability distribution for the 59 cases of northern

winds (not shown) is characterized by a high probability of very unstable flow both in the inflow (North of Crete) and in the low-speed zone. A small increase in the frequency of neutral winds in the jet zone is also noted. This result allows us to argue that the conclusions from Figure 12 can be generalized. The average stability for the 57 cases of westerly winds (not shown) has very unstable inflow to the west of Crete as in Figure 12. The high wind areas are associated with neutral or near-neutral stratification while the low wind zones along the coastlines to the north and south have very unstable stratification. This leads

us to conclude that the pattern in stability for westerly flow is similar between the single case (Figure 11) and average (Figure 10) except very near the coastline for low wind speeds.

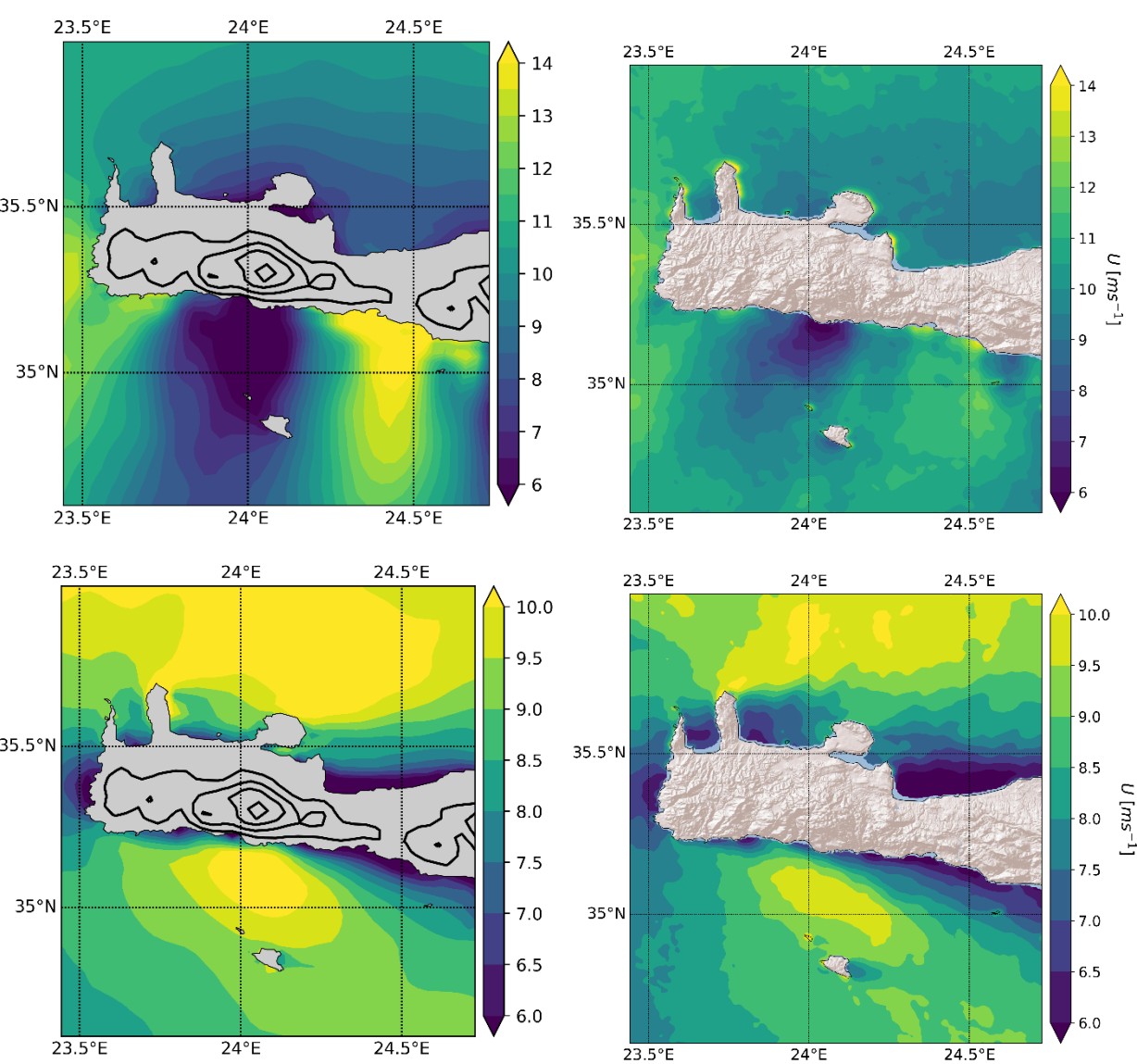

**Figure 10: Mean wind speeds (m s⁻¹) from WRF (left) and SAR (right) at 10 m. Top: Northerly flow based on 59 collocated cases from 2002 to 2018. Bottom: Westerly flow based on 57 collocated cases from 2002 to 2018.**

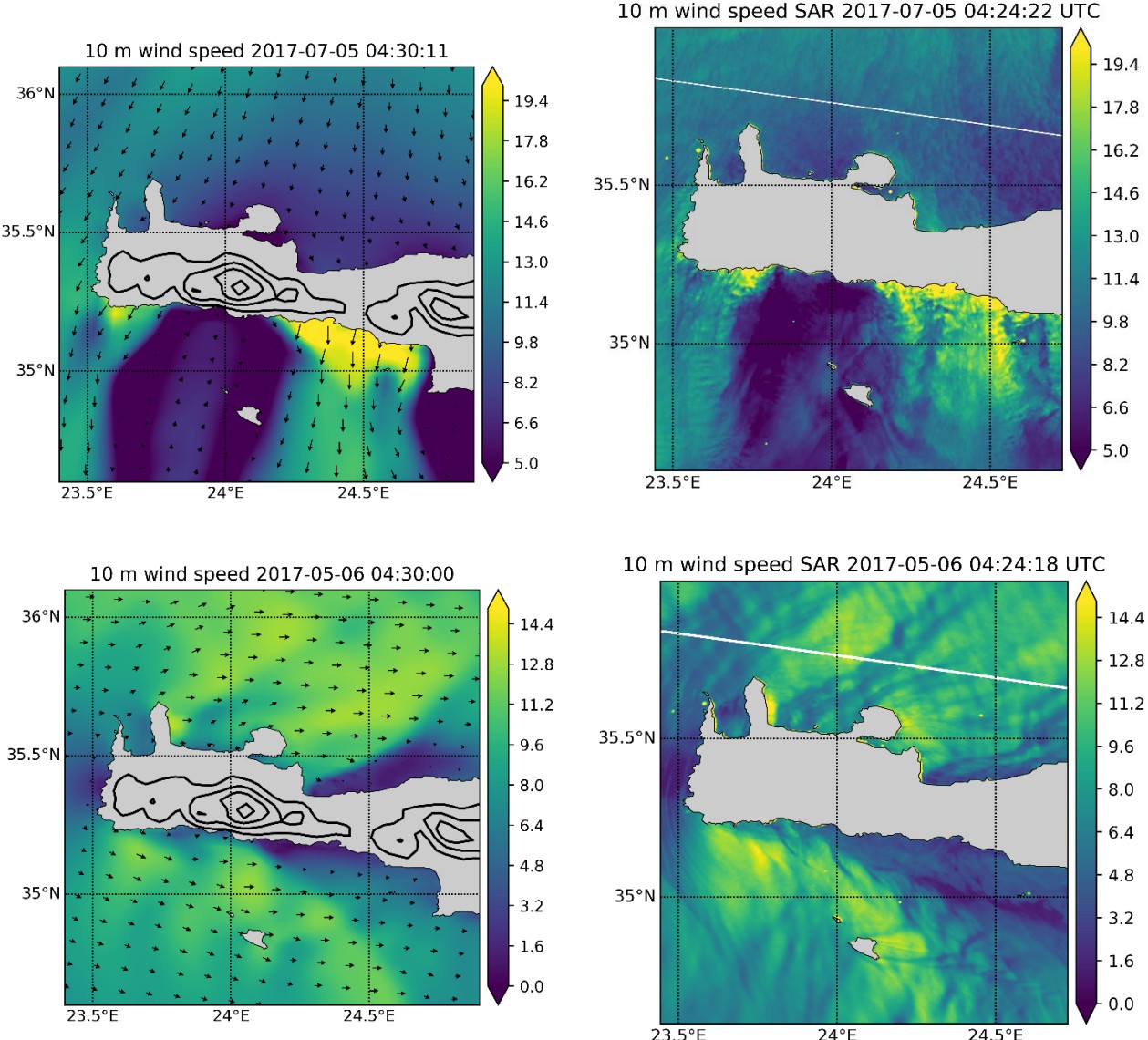

Figure 11: Wind speed (m s⁻¹) from WRF data (left) and SAR data (right). Top: Northerly flow 5 July 2017, 4:24 UTC. Bottom: Westerly flow 6 May 2017, 4:24 UTC. (the white lines in SAR panels are consecutive scenes borders).

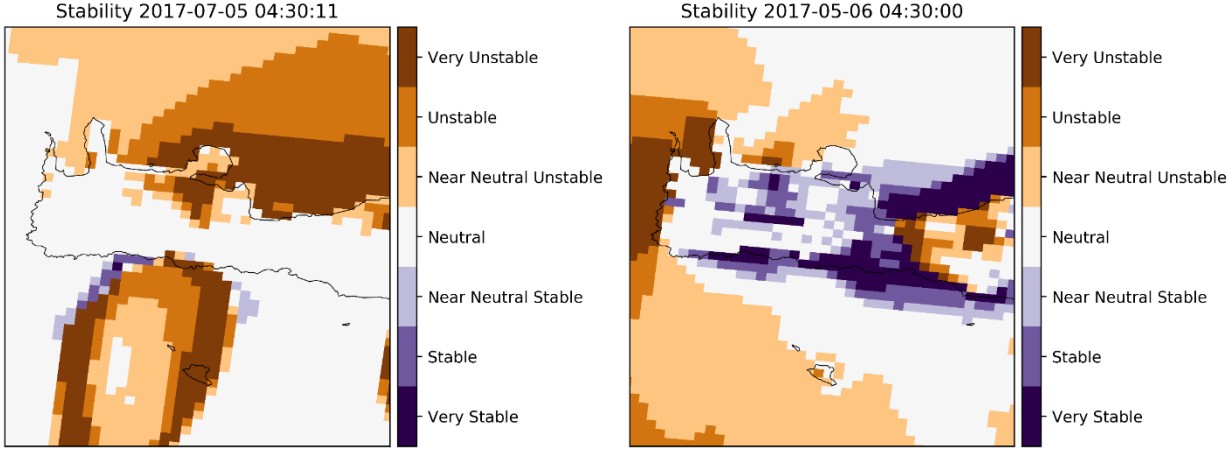

**Figure 12. Stability class following Gryning *et al.* (2007) in WRF for the cases with winds from the north (left) and from the west (right).**

## 6 Discussion

In wind resource mapping it is traditional to use hourly wind speed observations from one year (8760 samples) or ideally with higher temporal frequency and during more years from (tall) meteorological mast wind observations or wind profiling lidar. Offshore tall masts are few and thus, data are sparse. This stimulates research into atmospheric modelling and alternative observations, including satellite observations. At the onset of satellite data analysis for offshore wind resource mapping, few satellite scenes were available. Pioneering work (Barthelmie and Pryor, 2003; Pryor *et al.*, 2005) had focus on the number of samples relevant for assessing the mean wind speed, the Weibull scale and shape parameters and the energy density. Furthermore, the non-random sampling in time of sun-synchronous satellites that for ASCAT A/B are local times around 9:30 am/pm, Envisat around 10:30 am/pm and Sentinel-1 A/B around 06:00 am/pm potentially may bias the wind resource statistics, in the case of diurnal wind speed variations. The passive microwave wind observations with several more local observation times did not show much variation in diurnal cycle wind speeds in the central North Sea (Hasager *at al*., 2016) but near coastlines land-sea breezes prevail causing systematic diurnal wind speed variations.

Methods to deal with few satellite samples include the hybrid method (Badger *et al.*, 2010) and the gap-filling method during periods with lack of data due to sea ice (Doubrawa *et al.*, 2015). The adjustment for few samples and for uneven diurnal or seasonal sampling only makes sense to perform for local sites or regions (Ahsbahs *et al.*, 2019) rather than for the entire European Seas. In case meteorological observations are accessible, these can be useful for comparison and adjustment.

At the European scale, the SAR wind speed archive may be improved for future analysis, using the novel inter-calibration method proposed by Badger *et al.* (2019) and applied for SAR-based wind resource assessment along the US East Coast

(Ahsbahs *et al.*, 2019). The tendency in this inter-calibration is to decrease the SAR wind speeds. This obviously would make the comparison to both ASCAT and WRF agree better in the European Seas. Further validation of the offshore WRF winds with masts and lidar observations at around 100 m AMSL in the North Sea show smaller biases than those identified in Figure 5 and Figure 6 (Gonzalez-Rouco *et al.* 2019), which substantiates this hypothesis. It could furthermore be interesting to consider SAR and ASCAT inter-calibration such that coherent satellite data sets could be the foundation for further inter-

comparison to e.g. WRF model results. ASCAT and WRF test run comparisons (Karagali *et al.*, 2018a; 2018b) have proved valuable, as well as inter-comparison of WRF test runs and meteorological observations.

For planning of wind farms, statistics on wind speed and direction are crucial to agreeing on a central estimate of the long-term annual net energy production and for optimal design of turbine layout within the tender areas. ASCAT provides

observations of wind speed and wind direction, thus wind roses based on ASCAT are fully independent observations (e.g. Karagali *et al.*, 2018b). SAR only provides observations of wind speed, and for direction based upon the interpolated wind directions from global models (e.g. Badger *et al.*, 2010; Ahsbahs *et al.*, 2019). Thus, wind roses from SAR are mixed from satellite data and modelling. WRF provides modelled wind speeds and wind directions. ERA5 is a valuable data set, even though ERA5 resolves lesser spatial detail in offshore winds than the WRF production run, but ERA5 wind directions could

be an alternative to CFSR and GFS wind directions as input for SAR wind retrieval. It could potentially result in more homogenous SAR-based wind data set for the European Seas.

The opportunities for further investigations and analysis based on the New European Wind Atlas offshore are numerous. They include long-term wind speed and wind direction trends, future wind climate, comparison to various new wind data sources,

high fidelity modelling of winds, extreme winds, seasonal dependencies in winds, wind farm cluster effects between large offshore wind farms, wind energy production variability, new perspectives on marine boundary layer flows physics, processes and meteorological parameters, air-sea interactions, among other topics. It is the beginning of a new era in offshore wind energy research and applications.

**7 Conclusion**

The hitherto most comprehensive wind atlas for the European Seas has been published based on Envisat ASAR and Sentinel-1 A/B SAR satellite scenes, ASCAT A/B scatterometer satellite scenes and WRF mesoscale model production run results.

The WRF model covers 1989 - 2018 (30 years) with spatial grid spacing of 3 km and results every 30 minutes (in total 525,912 samples). The SAR wind archive covers from 2002 to 2018 with spatial resolution 2 km in total around 500 to 2,500 samples

during the years. The ASCAT wind archive covers from 2007 to 2018 with spatial resolution 12.5 km in total around 5,000 to 12,000 samples during the years.

Comparison results between SAR and WRF for the Crete case study reveal fine-scale flow structures in SAR not fully captured in WRF. However, overall ASCAT and WRF produce similar results of the mean wind speed across the European Seas at 465 100m height while SAR is positively biased. It is expected this bias may be diminished or removed using inter-calibration method for SAR.

**Acknowledgements**

The authors acknowledge the funding provided to the New European Wind Atlas project, in part funded by the European Commission's ERANET+, the Danish Energy Agency, and grants for supercomputers PRACE and EDDY. ASCAT data is 470 from the EUMETSAT, KNMI & the Copernicus CMEMS service. Envisat ASAR data is from ESA. Sentinel-1 data is from EC Copernicus. The SAR processing is based on the SAROPS software from JHU APL and NOAA. T.S. acknowledges the financial support of the project "Mathematical modelling of weather processes - development of methodology and applications for Latvia (1.1.1.2/VIAA/2/18/261)". The authors gratefully acknowledge the good collaboration with the WP3 partners of the NEWA project. Thank you to Rémi Gandoin and the two anonymous reviewers for their helpful assessment of the manuscript.

**Author contributions**

C.B.H. wrote the article and coordinated offshore wind atlas. A.N.H. coordinated the WRF modelling and T.S. assisted in WRF modelling and comparison. T.A. and T.S analysed the Crete case study. T.S did the stability analysis. I.K. analysed ASCAT. T.S. prepared the graphics. M.B. analysed SAR. J.M. coordinated NEWA experiments and project. All contributed 480 to discussion of and writing of the article.

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
