# Peer review of "Europe's offshore winds assessed from SAR, ASCAT and WRF"

_Wind Energy Science, 2019_

## Short Comment (SC1) · 30 Aug 2019

Dear authors,

What a nice paper, very interesting!

I have made some comments as I read it along, hopefully these can be useful.

All the very best
Rémi rga@c2wind.com

---Section 1---

Line 21: "in Europe". Possibly "European Union", https://windeurope.org/about-wind/statistics/european/wind-energy-in-europe-in-2018/#explore.

Line 24: "the total of all ocean-based industries". The metric used in (OECD, 2016) is "Gross Value Added".

---Section 2---

Lines 64-66: You could also mention http://projects.knmi.nl/knw/data/, https://www.dutchoffshorewindatlas.nl/ and http://marinedataexchange.co.uk/ItemDetails.aspx?id=4385.

Lines 88-89: You could also mention dual doppler RaDAR systems (BEACon project, see page 216 of https://books.google.dk/books?id=qfKZDwAAQBAJ).

Lines 93-95: " is valid in open-ocean and not near the coast". You could add a reference which helps the reader understand the reason for this (spatial averaging, water depth, fetch, or spatial/temporal heterogeneity/combination of these).

Lines 104: "due to the differences between land and sea influencing the atmospheric flow". It may be worth mentioning shortly if/when these differences are due to the physics of the model, or to the inputs (orography/roughness/SST ...) to the model (and/or a combination of these).

---Section 3---

General comment: it may be worth providing the reader with a reference to a document which explains the basic concepts of polarizations, maybe: https://earth.esa.int/handbooks/asar/CNTR1-1-5.html ?

Lines 125-126: "Wind product includes wind speed and wind direction at 10m height above sea level at spatial resolution of 12.5 km (de Kloe et al., 2017; CMEMS 2019)". What is the temporal resolution (i.e. time averaging and timestamping, for ex. 1-minute average reported 6-hourly)?

Table 2: may be worth adding the temporal resolution (see comment above).

Line 149: "the polarization ratio of Mouche et al. (2005) is selected". Not sure it is relevant here, but from the same author this newer study shows that in strong wind conditions, using VV or VH is not as good as a combination of both: https://www.researchgate.net/publication/318462216_Combined_Co-_and_Cross-Polarized_SAR_Measurements_Under_Extreme_Wind_Conditions. It may be worth providing a short argumentation on why and how the CMOD5.n algorithm has been used in combination with which polarization.

Line 152: "The SAR Ocean Products System (SAROPS) software". I could not find references to the software in (Monaldo, 2015), is there another reference which can help the reader? Could it be (Monaldo, 2005) MONALDO, F. M., 2005: ANSWRS: APL/NOAA SAR Wind Retrieval System-Software Documentation Version 3.0. Report: SRO-05-13. Johns Hopkins University/Applied Physics Laboratory, Laurel, MD 20723-6099, 36 pp.?

Line 161: "Wind turbines offshore operate at around 100m height". Maybe rephrase using rotor span and hub height. As a side note, this is also the height up to which the MOST can be used without making a too large error in stable conditions (see the works of A. Peña).

Line 161-165: I understand that you extrapolate the mean, annualised, Weibull-fitted wind speed distribution from 10 to 100 mMSL, is this correct? That, is, you do not extrapolate from 10 to 100 mMSL for each SAR scene.

Line 175: "compared to tall meteorological mast data". Also, LiDAR data (at the IJmuiden mast).

Line 177: "was setup". Typo: "was set up".

Line 182: "CORINE land cover data". It may be worth specifying which year.

Line 183-184: "The NOAH land surface model and icing WSM5 plus ice code and sum of cloud and ice humidity". The sentence is missing a verb.

Line 185: "61 vertical layers". It may be worth providing more details, in particular in the ABL.

---Section 4---

Figure 2: The Delta_U results (bottom right) could possibly be shown as relative differences instead of absolute differences. This may better reflect the magnitude of the stability correction when comparing two areas with different mean neutral-derived 100m wind speeds.

Lines 225-231: The discussion could include high-level explanation about mean sea surface- and air temperature spatial variations over Europe, i.e.: stable conditions occur when warm air flows over cold(er) water, etc., illustrated using ERA5 T2m-SST seasonal differences (here is a #nottobeproudof example: https://twitter.com/remi_wnd/status/1130428792386801664/photo/3).

Line 240 "wind sped". Typo.

Lines 244-248. The comparison of WRF with the sat data should here be understood as comparison between WRF and WRF-corrected sat data, right ? Would it make sense to show a comparison of sat data against 10m WRF neutral wind as well ?

Lines 246-247: "a consistent overestimation of winds from SAR". The word "overestimation" is best understood (#byme) when compared with measurements, maybe use "SAR data extrapolated using WRF long-term stability show a consistent negative bias compared with the WRF values"?

Line 247: "near the Dutch coastline". Also near the North-German coastline possibly? Maybe also mention that these artifacts are visible in Figure 2.

Lines 257-258: "These are attributed to the lack of ability in ERA5 to properly resolve the coastal atmospheric flow phenomena". In particular possibly due to the coarse spatial resolution of the model, see for instance in the Irish Sea: the Isle of Man is classified as sea in ERA5, same for Bornholm, thereby the oddity visible on the map shown here. See a subset of the mask here: https://www.dropbox.com/s/0zcpyujwd1sb984/ERA5_land_mask.JPG?dl=0.  As to the other differences, there are likely orography (and sometimes upwelling->stability) driven (Norway, South-East of France, Sicily, ...).

Line 265: "detail than". Missing "more".

Line 269: "ERA5 resolves scales around 150 km". Maybe add a reference. From the validation I have carried out, and others (https://www.dropbox.com/s/lx8aqky3cszuhbc/ERA5_spectra_vortex.JPG?dl=0), ERA5 seems to catch well the time variability. Offshore, the correlation of ERA5 against measurements is almost as large as with WRF (see slide 6 of http://c2wind.com/f/content/windeurope_wra_workshop_20190627_c2w_rev4.pdf and also https://www.linkedin.com/pulse/end-dynamic-mesoscale-models-wind-resource-assessment-bosch-i-mas/?trackingId=%2FhezBBaxCP0Qq68LxX8q2A%3D%3D).

Line 275: "overestimate the wind speeds". See my comment above, lines 246-247.

Figure 7: the blue and orange distributions are bimodal, what is the explanation for these? Do they correspond to different biases in stable and unstable conditions maybe?

---Section 5---

Line 285: "outline methodology". Missing "the".

Line 294: "WRF mean wind speeds". These are not corrected for stability, correct (i.e. not directly comparable to SAR-derived neutral wind)?

Figure 8: could you also add the mean annual or seasonal roses at selected locations around the island, for illustration? Maybe also the corresponding absolute mean difference between air and surface water temperature?

Lines 295-296: "A distinct jet south of the island is much more pronounced in the WRF data than in SAR". Could it be because these are Northerly (cold) winds flowing over warm water, and that the Figure compared 10m neutral wind from SAR to non-neutral WRF values? Also, there seems to be upwelling causing colder water to surface (like in the Gulf of Lion in France), see https://www.dropbox.com/s/2drn8k2oisss32i/WW_SST_crete_20170705.JPG?dl=0 .

Lines 301-302: "Koletsis et al. (2010) demonstrated the sensitivity of gap wind speeds in WRF to the changes in the elevation". Just a side note, I could not find the term "WRF" in this article (I used https://core.ac.uk/download/pdf/25867194.pdf).

Figure 9: maybe show the land/sea mask in ERA5 too.

Lines 335-336: "These features are present in the SAR as well, but much less pronounced". See my comment above, lines 295-296.

Figure 11: maybe add mean wind vectors too.

Figure 12: what where the atmospheric stability conditions around the island at these timestamps? SST maps are available from NASA Worldview, see https://www.dropbox.com/sh/dyfjamboyef73om/AABlCz99giL13q99mWUrbbkVa?dl=0 .

--- Section 6 ---

Line 373: "mast". And LiDAR too.

Line 399: "are crucial for optimal design of turbine layout within the 400 tender areas". From (#my) practitioner's perspective, these data are crucial to agreeing on a central estimate of the long-term annual net energy production, not so much on layout optimisation (i.e. where to place the turbines).

---

## Short Comment (SC2) · 3 Sep 2019

It is very valuable comments for our paper. We will take these points into consideration for the revision. Thank you very much indeed for taking time to help us improve the paper. Charlotte
* * *

---

## Referee Comment (RC1) · Anonymous Referee #1 · 27 Sep 2019

Line 22 Font size is smaller than the other. Line 49 "in prep", Paper in the preparation should not be referred. Other paper should be cited. Line 145 Table 2 should be Table 1 corresponding to the line 137. Line 248 "the dense population of large ships" will be "the congestion of large ships" or "the overcrowding of large ships". "Population" is not appropriate for ship conditions. Line 240 "sped" should be "speed". Line 258 What is the example of "coastal atmospheric flow phenomena"? Line 288 Figures 2 to 6 should be Figures 8 and 9. Figure 7 Vertical axis label should be inserted. Line 325 "decision flow chart" will be "flow chart" corresponding to Figure 10.

---

## Referee Comment (RC2) · Anonymous Referee #2 · 26 Oct 2019

Reviews corresponding to the article: "Europe's offshore winds assessed from SAR, ASCAT and WRF"

Authors: Charlotte B. Hasager, Andrea N. Hahmann, Tobias Ahsbahs, Ioanna Karagali, Tija Sile, Merete Badger, Jakob Mann

This paper is an important contribution to the wind energy community. Large amounts of data showing the wind resources in Europe's offshore regions is gathered and is compared to each other, making the advantages and disadvantages of each dataset clear. Finally, a case study is presented, indicating the spatial strength of SAR data and the advantages of WRF compared to ERA5. I recommend publication after the consideration of some minor comments!

**General comment:**

Case studies: You mentioned on P18L58, that stability can be crucial as well but that information about stability is not available. However, you could at least give some information according to the model whether the SST was much warmer than the 2-Temperature given in the ERA5 data or WRF model. Especially during northerly winds, it could be that the 2-Temperature is lower than the SST on the northern side of Crete but warmer on the southern side due to adiabatic warming. Additonally, it would be interesting to know, how compares the case study with northerly winds with a case study characterized by westerly winds in terms of SST – air temperature gradient.

**Specific comments:**

P1L14, space is missing: has 12.5 km and SARhas 2km → has 12.5 km and SAR has 2km

P6 Table 2: great job

P7L177-179: A reference to Fig. 1 would help a lot.

P7 Section 3.3: Mesoscale Modelling: Could mention the spacing of the vertical levels in the area of interest i.e. at common rotor height?

P7L178 "and later merged provide one unified atlas" → the reviewer suggests: „and later merged to provide one unified atlas"

P8 Fig. 2: The subplots are different in size

P12L258 A short note should be given here, that an example for a coastal atmospheric flow phenomenon will be investigated in section 5.

P14 Fig. 7: How was the distribution normalized; or what is the y-axis showing? It cannot be a probability in the classical sense as there are values greater than 1.

P16L321, P21393-394: Use a protected space between Figure and number of Figure

P18L349: What do you mean by wave maxima? Do you mean the hydrostatic jump, the location of wave breaking? Please be more precise, an annotation in Fig. 12 could help.

P21L393-395: The reviewer suggests to add: "than those identified in Figure 5" → "than those identified in Figure 5 and Figure 7"

---

## Author Comment (AC1) · 19 Nov 2019

Response to Referee #1 Thank you very much for your review with helpful suggestions for improvements. Line 22 Font size is smaller than the other. - Now it is font size 10. Line 49 "in prep", Paper in the preparation should not be referred. Other paper should be cited. - (Hahmann et al. in prep.) is deleted. Line 145 Table 2 should be Table 1 corresponding to the line 137. - Corrected Line 248 "the dense population of large ships" will be "the congestion of large ships" or "the overcrowding of large ships". "Population" is not appropriate for ship conditions. - Changed to "congestion" Line 240 "sped" should be "speed". - Corrected Line 258 What is the example of "coastal atmospheric flow phenomena"? - The sentence is expanded:…"such as land-sea breeze and flow intensification. See Beal et al. 2015 for further details on coastal

atmospheric flow phenomena observed in SAR. ". Line 288 Figures 2 to 6 should be Figures 8 and 9. - Corrected to "Figures 8, 9, 11 and 12" Figure 7 Vertical axis label should be inserted. - Done (the label is probability density distribution) Line 325 "decision flow chart" will be "flow chart" corresponding to Figure 10. - Deleted "decision"

 

Response to Referee #2 Thank you very much for your review with helpful suggestions for improvements. This paper is an important contribution to the wind energy community. Large amounts of data showing the wind resources in Europe's offshore regions is gathered and is compared to each other, making the advantages and disadvantages of each dataset clear. Finally, a case study is presented, indicating the spatial strength of SAR data and the advantages of WRF compared to ERA5. I recommend publication after the consideration of some minor comments! - Thank you for the positive words. General comment: Case studies: You mentioned on P18L58, that stability can be crucial as well but that information about stability is not available. However, you could at least give some information according to the model whether the SST was much warmer than the 2Temperature given in the ERA5 data or WRF model. Especially during northerly winds, it could be that the 2-Temperature is lower than the SST on the northern side of Crete but warmer on the southern side due to adiabatic warming. Additonally, it would be interesting to know, how compares the case study with northerly winds with a case study characterized by westerly winds in terms of SST – air temperature gradient. - We agree. We have chosen to elaborate on the stability directly instead of through SST. - So we have added (lines 377 to 397) plus new Figure 12.:

- The atmospheric stability from WRF has been investigated. We use the stability classes following Gryning et al. (2007) and map the stability at the time of the two cases shown in Figure 11. The results are presented in Figure 12.

- For the case with winds from the north, the inflow is very unstable at the northern shore of Crete, but in the high speed jet south of Crete the flow is neutral. In the low-speed zone south of Crete there is a complicated pattern of atmospheric stability – with a large zone of very unstable flow.

- For the case with winds from the west, the inflow (West of Crete) is again very unstable, and the wind speed maxima are again associated with neutral stability conditions, but this time in the low wind speed zones near the coastline north and south of Crete very stable stratification is found.

- The results in Figure 12 show a snapshot of stability in time, however, the focus on this paper is the climatological properties. Therefore, the next question arises – are these stability patterns typical. To answer this question, we analyze the stability patterns in the WRF dataset that coincides with SAR scenes (Figure 10). The stability distribution for the 59 cases of northern winds (not shown) is characterized by a high probability of very unstable flow both in the inflow (North of Crete) and in the low-speed zone. A small increase in the frequency of neutral winds in the jet zone is also noted. This result allows us to argue that the conclusions from Figure 12 can be generalized. The average stability for the 57 cases of westerly winds (not shown) has very unstable inflow to the west of Crete as in Figure 12. The high wind areas are associated with neutral or near-neutral stratification while the low wind zones along the coastlines to the north and south have very unstable stratification. This leads us to conclude that the pattern in stability for westerly flow is similar between the single case (Figure 11) and average (Figure 10) except very near the coastline for low wind speeds. Specific comments: P1L14, space is missing: has 12.5 km and SARhas 2km à has 12.5 km and SAR has 2km - Corrected P6 Table 2: great job - Thank you. We now write Table 1. P7L177-179: A reference to Fig. 1 would help a lot. - Done in line 177 P7 Section 3.3: Mesoscale Modelling: Could mention the spacing of the vertical levels in the area of interest i.e. at common rotor height? - We have added: There are 20 model levels below 1 km, the lowest levels are located at 5.6, 21.8, 40.4, 56.6, 72.8, 90.7, 113.2,

140.1, 170.7, 205.3, 244.5 m above ground level. P7L178 "and later merged provide one unified atlas" à the reviewer suggests: "and later merged to provide one unified atlas" - Done P8 Fig. 2: The subplots are different in size - Yes. Now all are same size. P12L258 A short note should be given here, that an example for a coastal atmospheric flow phenomenon will be investigated in section 5. - Done. P14 Fig. 7: How was the distribution normalized; or what is the y-axis showing? It cannot be a probability in the classical sense as there are values greater than 1. - We clarify in the Figure caption: The distributions are normalized so the area below the curves sums to one. P16L321, P21393-394: Use a protected space between Figure and number of Figure - Done P18L349: What do you mean by wave maxima? Do you mean the hydrostatic jump, the location of wave breaking? Please be more precise, an annotation in Fig. 12 could help. - We added: ..." the gravity wave amplitude maxima"... P21L393-395: The reviewer suggests to add: "than those identified in Figure 5" à "than those identified in Figure 5 and Figure 7" - Done  

Response to review from Rémi Gandion Thank you very much for your review with helpful suggestions for improvements. We very much appreciate your many great suggestions!

Line 21: "in Europe". Possibly "European Union", https://windeurope.org/aboutwind/statistics/european/wind-energy-in-europe-in-2018/#explore. - We think Europe is correct. Word "Union" is not mentioned in the reference. Norway is listed and is not member of the EU. Line 24: "the total of all ocean-based industries". The metric used in (OECD, 2016) is "Gross Value Added". - We changed from the total to "gross value added" —Section 2— Lines 64-66: You could also mention http://projects.knmi.nl/knw/data/, https://www.dutchoffshorewindatlas.nl/ and http://marinedataexchange.co.uk/ItemDetails.aspx?id=4385. - The two latter references are added - We add references KNMI (The Royal Dutch Meteorological Institute), Dutch Offshore Wind Atlas (DOWA) https://www.dutchoffshorewindatlas.nl/about-the-atlas, (last accessed 26 October 2019), 2019. The Crowne Estate, http://marinedataexchange.co.uk/ItemDetails.aspx?id=4385, (last accessed 26 October 2019), 2015. Lines 88-89: You could also mention dual doppler RaDAR systems (BEACon project, see page 216 of https://books.google.dk/books?id=qfKZDwAAQBAJ). - We added: "e.g. the Dual-Doppler radar (Nygaard and Newcombe, 2018; Valldecabres et al. 2019)" - We add references Nygaard, N.G. and Newcombe, A.C. Wake behind an offshore wind farm observed with dual-Doppler radars. J. Phys. Conf. Ser., 1037, 072008, 2018. Valldecabres, L., Nygaard, N.G., Vera-Tudela, L., Von Bremen, L. and Kühn, M. On the Use of Dual-Doppler Radar Measurements for Very Short-Term Wind Power Forecasts. Remote Sens., 10, 1701, 2018. Lines 93-95: " is valid in open-ocean and not near the coast". You could add a reference which helps the reader understand the reason for this (spatial averaging, water depth, fetch, or spatial/temporal heterogeneity/combination of these). - We did not add further reference as Hersbach is already listed. Empirical C-MOD GMF's (see refs. within Hersbach) are developed either based on buoy wind data and/or model wind data in the open ocean. Lines 104: "due to the differences between land and sea influencing the atmospheric flow". It may be worth mentioning shortly if/when these differences are due to the physics of the model, or to the inputs (orography/roughness/SST ...) to the model (and/or a combination of these). - We added. "The flow is more complex near the coastline than further offshore and fine scale structures may prevail such as land-sea breeze, not resolved by the model." —Section 3— General comment: it may be worth providing the reader with a reference to a document which explains the basic concepts of polarizations, maybe: https://earth.esa.int/handbooks/asar/CNTR1-1-5.html ? - We added: "This is for radar data in VV polarization. There is no GMF for HH data, therefore a conversion, the so-called polarization ratio linking the VV and HH data together needs to be applied before wind retrieval." Lines 125-126: "Wind product includes wind speed and wind direction at 10m height above sea level at spatial resolution of 12.5 km (de Kloe et al., 2017; CMEMS 2019)". What is the temporal resolution (i.e. time averaging and timestamping, for ex. 1-minute average reported 6hourly)? - We added: "Depending

on the area of interest, satellite overpass times can range from two to four per day, while the measurements are considered instantaneous (CMEMS-OSI-PUM-012-002)" Table 2: may be worth adding the temporal resolution (see comment above). - We thought of adding temporal resolution but it only makes sense for WRF (30 min). ASCAT and SAR both depend upon orbital paths. The graphics in Figure 2 (upper left panel) and Figure 3 (upper left panel) shows the total number. The number of overpasses depends upon latitude and longitude. We have written the approximate number in the text (Lines 215-219). - Line 149: "the polarization ratio of Mouche et al. (2005) is selected". Not sure it is relevant here, but from the same author this newer study shows that in strong wind conditions, using VV or VH is not as good as a combination of both: https://www.researchgate.net/publication/318462216_Combined_Co-_and_CrossPolarized_SAR_Measurements_Under_Extreme_Wind_Conditions. It may be worth providing a short argumentation on why and how the CMOD5.n algorithm has been used in combination with which polarization. - We are aware that the combination of co- and cross-polarization is superior for extreme winds. We only have co-polarized data and we do not focus on extreme winds in the current study. Line 152: "The SAR Ocean Products System (SAROPS) software". I could not find references to the software in (Monaldo, 2015), is there another reference which can help the reader? Could it be (Monaldo, 2005) MONALDO, F. M., 2005: ANSWRS: APL/NOAA SAR Wind Retrieval SystemSoftware Documentation Version 3.0. Report: SRO-05-13. Johns Hopkins University/Applied Physics Laboratory, Laurel, MD 20723-6099, 36 pp.? Thank you for pointing out we had included a wrong reference. We added: Reference "Monaldo, F.M., Li, X., Pichel, W.G. and Jackson, C.R.: Ocean wind speed climatology from spaceborne SAR imagery. Bull. Am. Meteorol. Soc. 95, 565–569. https://doi. org/10.1175/BAMS-D-12-00165.1, 2014. We deleted Reference Monaldo et al. 2015. Line 161: "Wind turbines offshore operate at around 100m height". Maybe rephrase using rotor span and hub height. As a side note, this is also the height up to which the MOST can be used without making a too large error in stable conditions (see the works of A. Peña). - We changed: "...have hub heights..." - We do not discuss the

rotor and e.g. wind shear and choose not to write about it here. We agree the topic is relevant but outside our scope. Line 161-165: I understand that you extrapolate the mean, annualised, Weibull-fitted wind speed distribution from 10 to 100 mMSL, is this correct? That, is, you do not extrapolate from 10 to 100 mMSL for each SAR scene. - Yes correct. We extrapolate mean stability statistics, not for individual scenes. Line 175: "compared to tall meteorological mast data". Also, LiDAR data (at the IJmuiden mast). - We added: "and lidar data" Line 177: "was setup". Typo: "was set up". - Done Line 182: "CORINE land cover data". It may be worth specifying which year. - We added: "Copernicus Land Monitoring Service, 2019)" Line 183-184: "The NOAH land surface model and icing WSM5 plus ice code and sum of cloud and ice humidity". The sentence is missing a verb. - We added: "are used." Line 185: "61 vertical layers". It may be worth providing more details, in particular in the ABL. - We added: "There are 20 model levels below 1 km, the lowest levels are located at 5.6, 21.8, 40.4, 56.6, 72.8, 90.7, 113.2, 140.1, 170.7, 205.3, 244.5 m above ground level". —Section 4—

Figure 2: The Delta_U results (bottom right) could possibly be shown as relative differences instead of absolute differences. This may better reflect the magnitude of the stability correction when comparing two areas with different mean neutral-derived 100m wind speeds. - We agree other ways of presenting is possible but we prefer the absolute numbers here. Lines 225-231: The discussion could include high-level explanation about mean sea surface- and air temperature spatial variations over Europe, i.e.: stable conditions occur when warm air flows over cold(er) water, etc., illustrated using ERA5 T2m-SST seasonal differences (here is a #nottobeproudof example: https://twitter.com/remi_wnd/status/1130428792386801664/photo/3). - We reformulated the sentence and added this reference instead that focuses on the stability in the Mediterranean Seas. - - We modified the text: "According to Kara et al. (2009) the overall stability in the Mediterranean Sea is slightly unstable." - The reference added is - Kara, A. B., Wallcraft, A. J. and Bourassa, M. A.: Optimizing surface winds using QuikSCAT measurements in the Mediterranean Sea during 2000–2006. Journal of Marine Systems, 8, S119-S131, 10.1016/j.jmarsys.2009.01.020, 72009. (We removed

another incomplete reference by Kara at al). We do not go deeper into the topic on stability over the entire European Seas but we have added our analysis on stability on the case study on Crete. (lines 377 to 399 plus new Figure 12)

Line 240 "wind sped". Typo. - Done Lines 244-248. The comparison of WRF with the sat data should here be understood as comparison between WRF and WRF-corrected sat data, right ? Would it make sense to show a comparison of sat data against 10m WRF neutral wind as well ? - Yes. We compare WRF and WRF-corrected satellite data. The reason for not also showing WRF and satellite data is due to 1) WRF performs generally better at 100 m and this is the height of interest; 2) Comparison of WRF and satellite at 10m have been presented in other papers and is beyond our scope. Lines 246-247: "a consistent overestimation of winds from SAR". The word "overestimation" is best understood (#byme) when compared with measurements, maybe use "SAR data extrapolated using WRF long-term stability show a consistent negative bias compared with the WRF values"? - We agree and have correct the formulation several places in the paper. Line 247: "near the Dutch coastline". Also near the North-German coastline possibly? Maybe also mention that these artifacts are visible in Figure 2. - Yes. We added: "(notice distinct yellow area in Figure 2 that without ships would be green, in other words, the high backscatter translate into falsely high wind speed)" - We do not think we see this near the German coastline. Lines 257-258: "These are attributed to the lack of ability in ERA5 to properly resolve the coastal atmospheric flow phenomena". In particular possibly due to the coarse spatial resolution of the model, see for instance in the Irish Sea: the Isle of Man is classified as sea in ERA5, same for Bornholm, thereby the oddity visible on the map shown here. See a subset of the mask here: https://www.dropbox.com/s/0zcpyujwd1sb984/ERA5_land_mask.JPG?dl=0. As to the other differences, there are likely orography (and sometimes upwelling->stability) driven (Norway, South-East of France, Sicily, ...). - We agree. We added: "ERA5 has a coarse spatial resolution omitting islands such as Bornholm and Isle of Man." Line 265: "detail than". Missing "more". - Done

[Figure]

Line 269: "ERA5 resolves scales around 150 km". Maybe add a reference. From the validation I have carried out, and others (https://www.dropbox.com/s/lx8aqky3cszuhbc/ERA5_spectra_vortex.JPG?dl=0), ERA5 seems to catch well the time variability. Offshore, the correlation of ERA5 against measurements is almost as large as with WRF (see slide 6 of http://c2wind.com/f/content/windeurope_wra_workshop_20190627_c2w_rev4.pdf and also https://www.linkedin.com/pulse/end-dynamic-mesoscale-models-wind-resource-assessment-bosch-imas/?trackingId=%2FhezBBaxCP0Qq68LxX8q2A%3D%3D).
-We added: "The use of structure functions and spectra from other numerical weather models have shown the effective model resolution to be of the order six times lower (Frehlich and Sharman, 2008)." We included the reference: Frehlich, R. and Sharman, R.: The use of structure functions and spectra from numerical model output to determine effective model resolution. Mon. Wea. Rev., 136, 1537–1553, 2008.

Line 275: "overestimate the wind speeds". See my comment above, lines 246-247. - Done Figure 7: the blue and orange distributions are bimodal, what is the explanation for these? Do they correspond to different biases in stable and unstable conditions maybe? - We added: "The long positive tails of ASCAT minus WRF and SAR minus WRF are explained by coastal winds in the Mediterranean Sea. ERA5 minus WRF has slightly positive bias and narrow range. All probability density distributions except AS-CAT minus SAR, show bi-modal distributions explained by the differences in area of the Northern Seas and Mediterranean Sea, see Figure 5." —Section 5— Line 285: "out-line methodology". Missing "the". - Done Line 294: "WRF mean wind speeds". These are not corrected for stability, correct (i.e. not directly comparable to SAR-derived neu-tral wind)? - True. WRF are not equivalent neutral winds. Figure 8: could you also add the mean annual or seasonal roses at selected locations around the island, for illustration? Maybe also the corresponding absolute mean difference between air and surface water temperature? - We calculated mean statistics for 360 degrees directions on stability from WRF at selected points but we find the results will take too many extra lines and choose not to go further into this. The main learnings from these are similar

to the analysis on stability we have included. Lines 295-296: "A distinct jet south of the island is much more pronounced in the WRF data than in SAR". Could it be because these are Northerly (cold) winds flowing over warm water, and that the Figure compared 10m neutral wind from SAR to non-neutral WRF values? Also, there seems to be upwelling causing colder water to surface (like in the Gulf of Lion in France), see https://www.dropbox.com/s/2drn8k2oisss32i/WW_SST_crete_20170705.JPG?dl=0. - We added: "Stronger winds may increase surface water mixing causing colder sea surface temperature (not shown)." Lines 301-302: "Koletsis et al. (2010) demonstrated the sensitivity of gap wind speeds in WRF to the changes in the elevation". Just a side note, I could not find the term "WRF" in this article (I used https://core.ac.uk/download/pdf/25867194.pdf). - We agree. We changed to "mesoscale model" instead of WRF. (The MM5 mesoscale model is predecessor to WRF).

Figure 9: maybe show the land/sea mask in ERA5 too. - We choose not to because ERA5 has another definition with percentage land/sea mask. Lines 335-336: "These features are present in the SAR as well, but much less pronounced". See my comment above, lines 295-296.

Figure 11: maybe add mean wind vectors too. - We prefer not to add as the information is given in the text.

Figure 12: what where the atmospheric stability conditions around the island at these timestamps? SST maps are available from NASA Worldview, see https://www.dropbox.com/sh/dyfjamboyef73om/AABlCz99giL13q99mWUrbbkVa?dl=0. -Please see our response to Referee 2 on the same question. — Section 6 —

Line 373: "mast". And LiDAR too. -We added:"or wind profiling lidar" Line 399: "are crucial for optimal design of turbine layout within the 400 tender areas". From (#my) practitioner's perspective, these data are crucial to agreeing on a central estimate of the long-term annual net energy production, not so much on layout optimisation (i.e.

where to place the turbines). - We added: "to agreeing on a central estimate of the long-term annual net energy production" - We did not delete the text on layout optimization as we think this is another perspective. *************** Changes we have done that do not relate directly to the Referee's comments and suggestion: We moved Figure 6 into Figure 5 (bottom right) and re-numbered subsequent Figures. We added ERA5 minus WRF probability density distribution on wind-speed difference error to the new Figure 6. We added: Metop-C was launched in 2018 although its data are not used in the present study. All We changed the link http://www.neweuropeanwindatlas.eu/ to https://map.neweuropeanwindatlas.eu

---

## Author Response (AR2)

Dear editor,

Thank you very much for the acceptance with minor revision.

Please find our response to the question:

We added line 172 to 175

The long-term stability is based on the Monin-Obukhov length and parameters on heat flux, air temperature and friction velocity from WRF as described in (Badger *et al.* 2016). The stability classes are from Kelly and Gryning (2010).

and we added the reference

Kelly, M., and S.-E. Gryning, 2010: Long-Term Mean Wind Profiles Based on Similarity Theory. Boundary-Layer Meteorol., 136, 377–390, doi:10.1007/s10546-010-9509-9

We also edited the affiliation for one co-author.

We added: Institute of Numerical Modelling,

On behalf of the co-authors

Best regards,

Charlotte

[revised manuscript text omitted]